# Review of Human Papillomavirus Vaccination Programs in United States Schools

**DOI:** 10.3390/vaccines13090894

**Published:** 2025-08-23

**Authors:** Cassandra Duran, Aditi Gupta, Lynda Aririguzo, Norma Castillo, Sanghamitra M. Misra

**Affiliations:** 1Department of Pediatrics, Baylor College of Medicine, 1 Baylor Plaza, Houston, TX 77030, USA; aditi.gupta@bcm.edu (A.G.); lynda.aririguzo@bcm.edu (L.A.); norma.castillo@bcm.edu (N.C.); smisra@bcm.edu (S.M.M.); 2Mobile Clinic Program, Texas Children’s Hospital, Baylor College of Medicine, Houston, TX 77054, USA

**Keywords:** Human Papillomavirus (HPV), HPV vaccine, adolescent vaccines, school-based vaccination programs

## Abstract

Background: School vaccination programs (school-based and school-located) that include the human papillomavirus (HPV) vaccine have been implemented throughout the United States since 2009. Methods: We conducted a review of school HPV vaccination programs in PUBMED, Google Scholar, Web of Science, Ovid, Medline, and Embase and included peer-reviewed studies originating in the United States that focused on any aspect of HPV school vaccination programs. Results: Our review yielded 47 articles that fell into several categories: (1) parent and child perceptions, (2) school nurse perceptions, (3) development, (4) implementation, (5) outcomes, and (6) barriers and facilitators of HPV vaccination programs in schools. Conclusions: School vaccination programs including the HPV vaccine have been implemented successfully all over the United States. Overall, nurse, parent, and student perceptions are positive, but there are various barriers to program success. Successes and failures of school HPV vaccination programs should be examined to develop best practices to sustain and expand these impactful programs.

## 1. Introduction

Human papillomavirus (HPV) is one of the most common sexually transmitted infections (STIs) worldwide and the most common STI in the United States [1]. Certain HPV genotypes are responsible for various cancers, including cervical, vaginal, vulvar, anal, and oropharyngeal cancers [2]. HPV vaccination has been a key strategy for preventing these cancers, with the introduction of vaccines such as Gardasil and Cervarix offering protection against the most high-risk strains of the virus [3]. Since the vaccines were first approved, several countries have implemented school-based or school-located vaccination programs as part of their public health initiatives [4]. These programs are seen as a critical tool in reaching adolescents in middle school (typically ages 11 to 14), who are the target age group for vaccination before potential exposure to the virus.

It is important to distinguish between various models of delivering vaccinations in or around school settings. *School-based vaccination programs* (*SBVPs*) are coordinated and administered directly on school grounds during school hours, typically by public health staff or school nurses, and are integrated into the school system as part of routine student health services. In contrast, *school-located vaccination programs* (*SLVPs*) may take place at the school site but are organized by external providers, such as health departments or mobile clinics, and are not formally embedded within the school’s infrastructure. These programs may occur during or after school hours and often require a parental presence or consent forms submitted in advance. *Extramural clinics* refer to off-site health care services located outside of traditional clinical settings, such as temporary setups in community centers or mobile units that serve school populations but are not physically located within schools but may be located outside of a school. Finally, *mass vaccination clinics* are large-scale, centralized efforts, often conducted during public health campaigns, to vaccinate a high volume of individuals in a short period of time. These may occur at schools. Understanding the distinctions among these delivery models is crucial for evaluating the access, uptake, and sustainability of HPV vaccination programs targeted at adolescents [4]. For simplicity in this paper, we will refer to all of the school-related vaccine programs as “*school vaccination programs*”.

The implementation of school vaccination programs that include the HPV vaccine has faced varied perceptions and challenges. Public attitudes toward HPV vaccination are influenced by factors such as cultural beliefs, a perceived link between the HPV vaccine and sexual activity, and parental concerns about safety and side effects [5,6]. In some communities, these perceptions create barriers to widespread vaccine uptake, which undermines a program’s potential public health impact. Additionally, school-based vaccination efforts are often shaped by local policies, health care infrastructure, and the involvement of school leadership, which can vary significantly from one school or state to another. In order to attempt to increase HPV vaccination rates among middle school children in the United States, all HPV vaccination programs in the United States should be reviewed and evaluated.

In this review, we examine the existing literature on perceptions of HPV vaccination in schools, as well as the design and effectiveness of school vaccination programs that include the HPV vaccine. We highlight strategies that have been successful in overcoming barriers to vaccination and identify areas where further improvements and advocacy are needed.

## 2. Methods

A thorough literature review of published articles on HPV school vaccination programs was completed in PUBMED, Google Scholar, Web of Science, Ovid, Medline, and Embase. We included peer-reviewed articles originating in the United States that focused on any aspect of HPV school vaccination programs. Search terms were “HPV vaccine school-based uptake programs” AND “school-based HPV vaccination programs.” The reference list for each peer-reviewed article was thoroughly reviewed in order to include all of the HPV school vaccination programs in the United States. We included studies involving children ages 9–18, in English language, no editorials, no letters, and no abstracts. We excluded articles on school vaccination programs that did not offer the HPV vaccine, did not take place in the US, or included college students.

## 3. Relevant Sections

Our review yielded 47 relevant articles that we categorized into groups: (1) parent and child perceptions, (2) school nurse perceptions, (3) development, (4) implementation, (5) outcomes, and (6) barriers and facilitators of HPV vaccination programs in schools.

### 3.1. Perceptions of HPV Vaccination Among Parents and Children

Parent and child comfort with vaccination programs in schools as a place to receive the HPV vaccination is generally high (50–81%) [7,8,9,10,11,12,13,14,15,16]. Acceptance of HPV vaccination at school depends on several factors. Students and families familiar with school-based health centers (pediatric clinics located inside schools) are more likely to feel comfortable receiving their HPV vaccine in the school setting [11,17,18]. Single parents, parents of uninsured children, and parents of children who either did not have a regular medical home or who had not been to the doctor in the last year are also more comfortable with receiving vaccinations in the school setting [7,19]. Parents of ethnic minorities, particularly Hispanic, and those with a lower education level are also more likely to agree to receiving vaccinations in school [12,19,20]. Acceptance could be related to an increased trust in school vaccination programs [16]. Families living in rural areas are more likely to choose locations outside the traditional medical home setting to get their child vaccinated [21]. An increased familiarity with the HPV vaccine and the intention to have their child vaccinated is also associated with increased comfort with receiving the vaccine in the school setting [7,13,14,20]. In households where both parents agree on the decision to vaccinate their child against HPV, the comfort with vaccination in the school setting is also higher [20]. Parents who want to vaccinate their child but face barriers in getting their child vaccinated are more comfortable with getting their child vaccinated in the school setting [7]. Parents who feel comfortable talking with their children about the HPV vaccine also feel more comfortable with alternative vaccination settings [7]. Convenience (e.g., not missing school, parent not missing work) is an important factor that both parents and adolescents considered when deciding whether to vaccinate at school [9,18,21]. Perceived adolescent comfort with the school setting also factors in parents’ comfort with allowing their child to be vaccinated in the school [21].

Parents who support their child getting vaccinated in school mention indirect benefits, such as increased adolescent responsibility, giving the adolescent time to learn about the vaccine themselves, and reinforcing the importance of the HPV vaccination [18]. Some parents consider the downstream effect of receiving vaccines in school such as the increased HPV vaccine uptake in the community [10]. Parents in one study cited “peer pressure” as an added benefit of vaccination programs in schools, in that seeing one of their friends get vaccinated against HPV may lead other kids to want the vaccine [9].

Both parents and adolescents have noted concerns with getting vaccinated at school. One of the most commonly discussed concerns was record keeping, specifically having all of the adolescent’s vaccination records in one place [10,13,18,19,21]. A lack of trust in school nurses as the person to administer the vaccination and a lack of trust in school-based health centers, in general, are reasons parents did not feel comfortable allowing their child to be vaccinated at school [8,9,18]. Many parents wanted to be present during the vaccination, which made them uncomfortable with their child receiving vaccinations in the school setting without the parent present [11,13,18,19].

Some adolescent perceptions of receiving vaccinations in school vary from those of their parents. Although adolescents, like their parents, associate HPV vaccination at school with increased responsibility and allowing them an opportunity to take care of their own health, they are concerned about fragmented medical care and would prefer to receive all of their medical care in one place [18]. Younger adolescents are concerned about the pain associated with getting the vaccine [18]. In a study looking specifically at male comfort with receiving the HPV vaccination in school, adolescent males cited significant embarrassment about getting vaccinated against HPV and their peers finding out [7]. However, the same study revealed that Hispanic male youth who live in urban areas and have an increased comfort with talking about HPV vaccination with their parents have an increased comfort with receiving the HPV vaccine in school [7]. Adolescents also report some confusion about the scope of practice of the school nurse and whether they could administer the HPV vaccine [8]. Concerns about vaccine safety and side effects were also cited by adolescents [8]. Older adolescents report higher comfort in receiving the HPV vaccine outside of the traditional medical setting than younger adolescents [14,17].

The HPV vaccine is not mandatory for school entry in the US. There are mixed perceptions of whether it should be required. Lower income parents or those whose children are uninsured or have Medicaid are more likely to agree that HPV vaccination should be required for school [22]. Parents who believe that HPV vaccination reduces the risk of cervical cancer and those who have a personal history of HPV are more likely to agree that HPV vaccines should be mandated in schools [22]. One study found that 43% of parents support mandatory HPV vaccination in schools [15]. Concerns about adverse effects, the perceived newness of the vaccine, and concerns about infringing on patients’ rights contribute to this relatively low agreement [9,15]. Concerns about parental autonomy also result in not supporting a mandatory HPV school policy [9]. Reasons to support HPV vaccine school entry requirements included improvements in public health as well as doing the right thing [9].

Comfort with receiving HPV vaccination in schools as compared to comfort with receiving other adolescent vaccinations including TdaP, MCV4, and Influenza varies. Willingness to consent for HPV vaccination ranges from 27 to 53%, which is lower than the willingness to consent for other adolescent vaccinations (57–67% for Influenza, 41–72% for TdaP, 35–71% for Meningitis) [12,19,23]. In one private school study, the HPV vaccine was the least frequently chosen vaccine (6%) in their school-located immunization program compared with 53% for Influenza, 28% Tdap, and 23% Meningitis [12].

### 3.2. Perception of HPV Vaccination Among School Nurses

School nurses are ideal partners for promoting HPV vaccination since they monitor student immunization compliance, specifically in underserved areas [24]. School nurses feel well suited to promote HPV, especially since discussing HPV is easily integrated into their existing work, and they find it to be valuable for increasing vaccination awareness and student compliance/vaccination rates. Nurses note that they can benefit from support, like multilingual educational materials [25]. School nurses generally support HPV vaccination for students in schools because it provides an opportunity for improved access, convenience, and cost-effectiveness, but they identify major logistical challenges, especially around parental consent, event staffing needs, and time constraints [8].

### 3.3. Development of HPV Vaccination Program in Schools

The development of targeted and successful vaccination programs in schools is determined by several factors, such as the age range of patients, cost (time and resources), and long-term impact of preventing and reducing HPV-related cancer rates [26]. Creating a vaccination program generally begins with responding to a community need, procuring funding, and creating partnerships with key stakeholders that allow access to schools and to families in schools [27]. Programs have been developed by first gauging the interest and perception of the need for HPV vaccines in schools through surveys and interviews of school leaders [27]. Obtaining the school staff’s perspectives via in-person interviews rather than phone calls not only aids in building a rapport and trust with the team but also helps develop shared goals for the students [28,29]. A strong partnership with school staff is essential for program longevity, since school staff are representatives for parents and can communicate as liaisons for the program, including making follow up phone calls, securing sites, obtaining records, collecting consent packets, and educating families [30,31]. Providing a written proposal (including information like HPV prevalence, vaccination rates, and the program plan itself) to the school leadership can lay the foundation of the program [27]. The program plan can include a project timeline, a recruitment outline, a consent form distribution and tracking plan, verifying vaccine eligibility, obtaining and administering the vaccine, the documentation of potential adverse reactions and concerns, and the documentation of vaccines administered in clinics on student records [31]. The participation from school staff and leadership strengthens if there is a shared interest for the students and a full transparency of the program goals from the start [30,31]. Once a program is established, members to consider adding to the team include project managers, clerical support, health educators, school nurses, and staff to vaccinate.

Program developers should consider existing laws, potential legal barriers, and specific school needs when developing their intervention [32]. Programs should also consider including all school-mandated vaccines along with the HPV vaccine, school staff availability (hours and assigned roles), and the education literacy level for promotional and communication material created for families. Tailoring the program to school needs, aligning with school goals, and not increasing the burden on school staff are helpful for a program to run smoothly [32]. Programs should consider how best to minimize and simplify staff and parent procedures and workloads. Equipping a school nurse with timely program information and valuable vaccine education aids them in their role as a community leader and helps them support parents with confidence and evidence-based knowledge. This is feasible through free or low-cost state health resources and continuing education [29].

One important way to increase parent trust and program participation is to appropriately tailor the literacy level of educational materials about the HPV vaccine and its importance [32]. Effective strategies include a parent-focused approach when promoting the program, providing comprehensive vaccine care along with the HPV vaccine (include additional adolescent vaccines like MCV and Tdap), and providing ongoing education for school nurses [29,33]. The language and tone regarding vaccine importance should stem from a compassionate viewpoint, highlighting the prevention of cancer and emphasizing the equal importance of the HPV vaccine in comparison to other adolescent vaccines. Regarding consent forms, options that improve participation rates include developing forms with “opt-in” rather than “opt-out” options [33]. Programs should highlight advantages of the program, such as fewer doctor appointments, less parent time off work, and less time out of school for the child. Another point of parent buy-in is the age of the child. High school students are considered more accessible than middle school students for these school programs [31]. Access to HPV vaccine education is important for parents and school nurses, and overall open communication regarding the program improves trust and promotes program continuity [29,33].

Setting realistic expectations and considering challenges are important quality improvement strategies for program development. Challenges include billing procedures, insurance verification, eligibility for services, and often time-consuming grant writing [34]. Acknowledging and properly planning for these factors can positively impact program longevity.

### 3.4. Implementation of HPV Vaccination Programs in Schools

HPV vaccination in schools is generally implemented through a school-based health center (SBHC)/school-based vaccination program, school-located vaccination program, or extramural program [30,35,36]. SBHCs are clinics located within schools that provide comprehensive medical care to students [30]. School-located vaccination clinics and extramural programs are usually partnerships between schools and community organizations (such as local health departments and non-governmental organizations) to deliver vaccines to schools without SBHCs [30,35].

SBHCs, operating in approximately 6% of schools in the US, generally bill public and private third-party payers for vaccines [35,37]. As comprehensive medical clinics, most have systems in place to deliver and track vaccinations [35,37]. To receive care at an SBHC, parents must sign a general consent form that covers the range of services provided by the SBHC, including vaccines, but many also require additional vaccine-specific consent [38]. Most SBHCs care for adolescent patients in absence of their parents [37]. Some states may allow minors to self-consent for services that prevent STIs, including the HPV vaccine; still, SBHCs must comply with the school policy regarding self-consent for HPV vaccines [38].

School-located vaccination clinics and extramural programs require coordination among partners [27,30,31,34,35,36]. The school district and school leadership may be involved early in the planning process/project development, perhaps as early as the grant-writing phase of planning, as it may improve the program implementation [27,30,35]. While programs can choose to offer only HPV vaccines, targeting all recommended vaccines is common and can increase acceptability and HPV vaccine uptake [27,30,36]. School-located vaccination clinics and extramural programs often include educational components targeting not only parents but also students, teachers, school administration, other school staff, and health care providers [27,30,34,36]. To promote and deliver vaccinations, programs can take advantage of school events where parents and guardians are likely to attend, such as registration events, orientations, parent–teacher meeting days, or school festivals [30,31]. Promotions through Parent–Teacher Association (PTA) meetings, social media, the radio, school automated call systems, community advocates, and grant-funded student incentives are also effective options [31,36].

All school vaccination programs must obtain parental consent prior to vaccination. Printed consent forms can be sent home with students, and students can then return consent forms to teachers, school nurses, or other school representatives, or the program can provide stamped envelopes for parents to mail back the consent forms [27,30,34]. Some programs use electronic consent procedures. Programs can also obtain consent at school events during promotion, which allows parents to talk to staff directly if they have questions [30,34]. Many programs also utilize reminder systems for parents, for both the initiation and completion of the vaccine series; examples include automated voicemails from schools, school newsletters, postcards, and letters [30,31,34,36]. Programs should work with school officials to determine if parents can be absent during vaccination events. Some schools require parents to be present at the series initiation only, while others require parents to be present at all vaccinations at school despite prior written consent [30].

Consent packets commonly include a consent form, health insurance information, health questionnaires, and student immunization records [30]. School personnel often review consent packets prior to the partner vaccinator’s arrival, as the Family Educational Rights and Privacy Act (FERPA) restricts the release of certain student information to third parties without consent [34]. To decrease the burden on existing school personnel, some programs hire paraprofessionals to carry out consent processing and other FERPA-restricted school activities [34].

Programs must coordinate with schools to determine the preferred time of the school day for vaccinations. They may choose to vaccinate before, during, or after school or at other school events where parents may be present [30,36]. Similarly, programs should work with school personnel (e.g., school nurses) to select the appropriate clinic location within the school and to coordinate the flow of patients [30,36]. Partner vaccinators review health insurance information, health questionnaires, and vaccine histories before vaccinating [34]. Clinics must have a means to transport vaccines at their appropriate temperatures, such as portable vaccine refrigerators [34]. School-located vaccination clinics generally follow the same clinical process: confirm consent prior to vaccination, vaccinate, document vaccination in internal records, and provide students with written documentation of vaccination [30,34,35,36]. Programs should confirm the vaccination status of each student prior to vaccination. Obtaining access to all school vaccination records is helpful and requires school support [30]. Programs usually do not charge a fee or co-pay to students; programs usually receive vaccines without cost through the Vaccines for Children (VFC) program, though some programs have lead partners who obtain private-stock vaccines and can bill public and private third-party payers [30,34,35].

### 3.5. Outcomes of HPV Vaccination Programs in Schools

A variety of studies have been published analyzing the effectiveness and impact of HPV vaccination in schools. Although studies focus on different outcome measures, all the data is useful in understanding the effect of these programs. Table 1 outlines 13 articles that focus on the outcomes of HPV vaccination programs in schools.

There are important points to note from these studies. Parents in lower income settings are willing to utilize HPV vaccination programs in schools to protect their adolescent children from HPV [43]. However, analyzing HPV vaccination trends, there has been a significant increase in the use of private clinical facilities and a decrease in the use of public and alternate facilities from 2014 to 2020 [47]. In one study of 74,645 participants, 65.9% reported that they received the HPV vaccine in private facilities, and 91.9% reported receiving the HPV recommendation from a health care professional [47]. Only 2.7% of the 74,645 participants received their HPV vaccination in an alternate facility, such as a school-based health center, pharmacy, or other type of facility [47]. A comparison study analyzed the effectiveness of community-based HPV-related education and on-site school-based vaccination versus community-based HPV education only [42]. Students enrolled in the intervention school were >3.6 times more likely to get the HPV vaccine than students enrolled in the comparison schools. There was a 34% increase in HPV vaccine initiation at the intervention schools versus a 13% increase at the comparison schools (*p*-value < 0.001). As for newly completed vaccine series completion rates, there was a 20% increase in the intervention school versus a 6% increase in the comparison school [42]. This study provides evidence that school-based vaccination clinics along with education increases HPV vaccine rates more than education alone [42].

One study reviewed HPV vaccine doses in the immunization registry for adolescents born during 1995–2000 in Seattle, Washington, to record how many used school-based health centers for any of the HPV vaccine doses. Of the 12,676 females who received the HPV vaccine, 1186 (9.4%) received one or more doses from an M-SBHC, and 511 (4%) received one or more doses from an M-SBHC [44]. In a school-based vaccination program with economically disadvantaged students in the Rio Grande Valley, Texas, 884 females and 882 males received at least one or more doses of the HPV vaccine through a school-based vaccination program [46]. Many of the middle school students received the HPV vaccine at the age of 11 (39.5%) or 12 (30.5%). The total HPV compliance rate was 59.7%. Students who initiated the HPV vaccine at age 11 were more likely to complete the series compared to those who initiated it at age 12 or older [46]. Overall, school vaccination programs may be a small percentage of total HPV vaccinations administered in many areas of the US, but they likely fill a gap and help many adolescents receive an important vaccine.

## 4. Discussions

Implementing HPV vaccination programs in schools often results in a variety of logistical challenges, and many of the challenges are uniquely different for each program. Various factors affect success, such as the school leadership buy-in, consent procedure difficulties (parents not receiving the consent forms and children not returning the forms or returning incomplete forms), the high cost for private-stock vaccines for those ineligible for Vaccines for Children vaccines, billing/collection complications (if applicable), the appropriate transport of vaccines, coordination with school nurses, difficulty accessing statewide immunization registry data, and student schedules allowing vaccination during the school day [4,39,47,48].

Possible strategies to increase HPV vaccine uptake include student incentives, parent reminders, and obtaining consent from parents when at the school [48]. A New York City school-based health center group providing school-based vaccinations conducted a quality improvement process after identifying low HPV vaccination initiation and completion rates (76% and 43% for high schools and 81% and 45% for middle schools). Schools that allowed adolescents to consent for themselves (permissible in New York) or did not require parents to complete the vaccine consent in addition to the annual medical consent had higher baseline HPV vaccination rates. Difficulties with scheduling vaccine appointments and high no-show rates were identified as areas for potential QI, and automatic reminder calls were instituted. After implementation, in participating high schools, a significant increase in HPV vaccine initiation (2.9%, *p* < 0.01) and series completion (2.7%, *p* < 0.05) was seen, but no statistically significant difference was seen in middle schools [38]. Each program must identify their specific barriers and then systematically address the issues in collaboration with all stakeholders. And by focusing on the facilitators, programs can sustain, and even expand, their programs.

## 5. Conclusions

There is significant evidence to support HPV vaccination programs in schools. In general, HPV vaccination programs in schools improve HPV vaccination rates. An increase in HPV knowledge and awareness was also noted with the use of HPV vaccination programs in schools. The convenience of these programs is noted by parents and children, and school leadership is generally appreciative of the collaboration to protect their students from HPV disease and potential cancers. Despite the overall effectiveness of these programs, the success of HPV vaccination programs in schools varies widely by program.

## 6. Future Directions

Strategically focusing on barriers and facilitators of HPV vaccination programs can greatly affect the outcomes of these programs. To continue improving HPV vaccination rates across the country, vaccination programs in schools should be promoted and expanded.

## Figures and Tables

**Table 1 vaccines-13-00894-t001:** Outcomes of HPV vaccination programs in schools.

Author (year)	Study Design	Sample	Evaluation Focus	Outcomes
Daley et al. (2014) [39]	Cluster-randomized controlled trial	6th–8th grade students at seven Denver public schools during 2010–2011 school year *n* = 3144	Vaccine administration and purchase costs were compared in vaccination program versus reimbursement by insurers	Estimated vaccine cost of USD 23.98 per vaccine in vaccination program versus USD 11.83 to USD 29.77 per vaccine reimbursement by insurance
Shah et al. (2020) [35]	Quality improvement project	Two Los Angeles area school-based health centers	Compared missed opportunities for HPV vaccine in school-based health centers after a multicomponent intervention	Missed HPV vaccine opportunities decreased from 82.3% to 46.1% during intervention period
Gold et al. (2011) [40]	Observational study	19 school-based health centers in Oregon	Evaluated completion of HPV vaccine series	51% completed HPV series by December 2008
Federico et al. (2010) [41]	Retrospective analysis	Immunization registry for patients aged 12–18 years old	Compared completion rates at school-based health centers vs. community health centers	School-based health centers had significantly higher immunization (Hepatitis B, Tdap, IPV, Varicella, MMR, and HPV) series completion rates compared to community health centers
Kaul et al. (2019) [42]	Quasi-experimental study	2307 Rio Grande City Consolidated Independent School District (RGCISD) middle school students from 3 schools	Comparison study of community-based HPV-related education and onsite school-based vaccination versus community-based HPV education	Students with on-site vaccination events and community-based education had higher HPV vaccination rates than those who only received community-based education
Middleman et al. (2016) [43]	Cluster—randomized controlled trial	Large Texas school district—449 students participated in the fall and 161 students participated in the spring	Evaluated HPV vaccine uptake in a school-located vaccination program	86% of students who were immunized received the HPV vaccine
Munn et al. (2019) [44]	Retrospective study	Immunization registry data of adolescents in Seattle, Washington, born between 1995 and 2000 who received an HPV vaccine	Identified adolescents who used a school-based health center for any HPV vaccine dose	Adolescent SBHC users had higher odds of completing the HPV vaccine series—37% higher odds for females and 45% higher odds for males
Rodriguez et al. (2022) [45]	Quasi-experimental study	*n* = 6481 middle school students	Assessed HPV vaccinations rates during the COVID-19 Pandemic	HPV vaccine initiation and completion rates increased 1.29-fold and 1.47-fold with a school-based vaccination program
Rodriguez et al. (2022) [46]	Quasi-experimental study	*n* = 1766 middle school students	Compared HPV vaccination rates by age of initiation	Initiating HPV vaccine at age 11 or younger increased completion series rates with a school-based vaccination program
Shaly et al. (2015) [34]	Descriptive study	3 clinics at 7 middle school–8th grade schools	Implementation of a school-located vaccination program	SLV program offers an alternative approach for providing vaccinations
Vanderpool et al. (2015) [31]	Descriptive study	2 high schools during the 2012-2013 academic year *n* = 935	Implementation and evaluation of a school-based HPV vaccination program	88% of students initiating the vaccine series completed the series; 315 initiated the HPV vaccine series; 276 completed the entire three-dose series
Stubbs et al. (2014) [27]	Cluster – randomized controlled trials	Temporary clinics in middle school in 2009–2010 *n* = 7916	Evaluation of a school-located HPV vaccination clinic	HPV initiation was higher in girls attending host school
White et al. (2024) [47]	Cross sectional analysis	National Immunization Survey-Teen data from 2014 to 2020 *n* = 74,645	Described trends and factors associated with HPV vaccine uptake	Odds of receiving HPV vaccine at a public facility vs. a private facility increased two times for adolescents living below poverty

## Data Availability

The original contributions presented in this study are included in the article. Further inquiries can be directed to the corresponding author(s).

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
