# Peer review of "Review of Human Papillomavirus Vaccination Programs in United States Schools"

_vaccines, 2025, doi:10.3390/vaccines13090894_

Round 1
Reviewer 1 Report
Comments and Suggestions for Authors
Dear authors,
Thank you for submitting the manuscript titled "Scoping Review of Human Papillomavirus Vaccination Programs in United States Schools" for "Vaccines" journal, and thus, for giving me the opportunity to read it.
After careful reviewing, I am confirming the interest of the manuscript and the importance of the subject of vaccination in US schools, given the fact that HPV is known as a public health concern in USA and one of the most STI actually, it is also playing a role in cancer risks, making it a hot topic.
Nonetheless, it is with some revisions that I recommend the manuscript for further process and publication in the future, kindly find my comments below:
- Line 4-5, please write only your names, do not put MD, MSN...etc.
- Line 39-56, maybe it is better to support this paragraph and different vaccination programs by some references
- In the end of the introduction section, it is not very clear why you have chosen to conduct this review, I think the problematic is a little lacking. In addition, it is better for the readers if you clearly emphasize the aim of your study.
- Line 70-71, please avoid using exactly the same sentence in the methodology section and the abstract section.
- I would like to know what were the keywords used by the authors to conduct this search ?
- What were the main criteria that helped you to select these inclusion and exclusion criteria ?
- Please homogenize the way you write the results of the P value, (eg. Line 309 .0.01 or line 347 0.01) choose to put or delete the zero.
- Honestly, the results were well discussed and the writing was very well conducted in general, I would like to congratulate the authors for this work.
- I think the conclusion section is a little weak, please try to extend it a little and try also to give more recommendations.
- Please try to include more recent studies, specifically from 2024-2025, and more specifically to support general statements (not necessarily those included in the search, but those used in the introduction section as an example).
I wish the authors good luck.
Author Response
- Line 4-5, please write only your names, do not put MD, MSN...etc. - done
- Line 39-56, maybe it is better to support this paragraph and different vaccination programs by some references – cited
- In the end of the introduction section, it is not very clear why you have chosen to conduct this review, I think the problematic is a little lacking. In addition, it is better for the readers if you clearly emphasize the aim of your study.- corrected
- Line 70-71, please avoid using exactly the same sentence in the methodology section and the abstract section. - corrected
- I would like to know what were the keywords used by the authors to conduct this search ? – Added to the methods section
- What were the main criteria that helped you to select these inclusion and exclusion criteria ? – The main criteria that determined the inclusion and exclusion criteria was based on the need to create a narrative review of HPV school based vaccination programs in the United States given a gap in the United States literature.
- Please homogenize the way you write the results of the P value, (eg. Line 309 .0.01 or line 347 0.01) choose to put or delete the zero.- corrected
- Honestly, the results were well discussed and the writing was very well conducted in general, I would like to congratulate the authors for this work.- YAY! J Thank you so much!
- I think the conclusion section is a little weak, please try to extend it a little and try also to give more recommendations. – See manuscript
- Please try to include more recent studies, specifically from 2024-2025, and more specifically to support general statements (not necessarily those included in the search, but those used in the introduction section as an example). – no additional studies were found.

Reviewer 2 Report
Comments and Suggestions for Authors
Congratulations on the excellent overview of the School Vaccination Programs in the United States.
I would like to offer a few small corrections:
-
In the introduction, it would be important to mention the age groups that could be included in the school-based programs.
-
Lines 31 and 34: Instead of strains, you should use genotypes.
-
Lines 317–318: In both options, you have SBHC. According to the bibliography, the second option should be M-SBHC.
-
You could widen Table 1 slightly to make it more visually pleasant and easier to read.
Author Response
Congratulations on the excellent overview of the School Vaccination Programs in the United States.
I would like to offer a few small corrections:
- In the introduction, it would be important to mention the age groups that could be included in the school-based programs. - see manuscript
- Lines 31 and 34: Instead of strains, you should use genotypes. - corrected
- Lines 317–318: In both options, you have SBHC. According to the bibliography, the second option should be M-SBHC. - corrected
- You could widen Table 1 slightly to make it more visually pleasant and easier to read. - corrected
